# A Miniaturized Arc Shaped Near Isotropic Self-Complementary Antenna for Spectrum Sensing Applications

**DOI:** 10.3390/s23020927

**Published:** 2023-01-13

**Authors:** Ubaid Ur Rahman Qureshi, Shahid Basir, Fazal Subhan, Syed Agha Hassnain Mohsan, Muhammad Asghar Khan, Mohamed Marey, Hala Mostafa

**Affiliations:** 1School of Optics and Photonics, Beijing Institute of Technology, Beijing 100081, China; 2School of Engineering & Applied Sciences, ISRA University, Islamabad 11622, Pakistan; 3Ocean College, Zhejiang University, Zheda Road 1, Zhoushan 316021, China; 4Hamdard Institute of Engineering and Technology, Hamdard University, Islamabad 11622, Pakistan; 5Smart Systems Engineering Laboratory, College of Engineering, Prince Sultan University, Rafha Street, P.O. Box 66833, Riyadh 11586, Saudi Arabia; 6Department of Information Technology, College of Computer and Information Sciences, Princess Nourah bint Abdulrahman University, P.O. Box 84428, Riyadh 11671, Saudi Arabia

**Keywords:** isotropic pattern, self-complementary, solid angle, spectrum sensing, IoFT, drones, 5G, ultra-wideband (UWB), cognitive radio (CR), UAV

## Abstract

This paper presents the design of an arc-shaped near-isotropic self-complementary antenna for spectrum sensing application. An arc-shaped dipole with horizontal and vertical arms is used to achieve a near isotropic radiation pattern. The radiation pattern improved by adjusting the horizontal and vertical arm lengths. Simulated and experimental results show that the proposed antenna has an impedance bandwidth of 146% (2.4–18.4 GHz) for VSWR ≤ 2 with a good radiation pattern. In order to quantify the antenna performance, antenna gain variation, bandwidth, efficiency, and size have been compared with previously reported designs. It is shown that the proposed arc-shaped antenna can achieve nearly isotropic radiation patterns with a maximum radiation efficiency of 92%. The isotropic performance of the antenna has been characterized by observing the radiation pattern and solid angle. The FR4 substrate is used as a dielectric with relative permittivity 4.4 and loss tangent of 0.02. ($\varepsilon_{r}$ = 4.4, *h* = 1.6 mm) The simulated and measured results are in good comparison, and the proposed design is a suitable candidate for spectrum sensing.

## 1. Introduction

### 1.1. Background and Related Work

The radio frequency spectrum is a critical resource that must be used effectively and efficiently. In order to minimise unwanted interference between different frequency bands, the Federal Communications Commission (FCC) controls the allotment of RF spectrum and distributes it amongst the licensed users WiMax, WLAN, and X-band. The FCC allocated band for IEEE 802.16 WiMAX system is (3.3–3.9 GHz) and the IEEE 802.11a wireless local area network (WLAN) system (5.15–5.825 GHz) and the X band is (8–9 GHz). This results in a waste of the radio frequency spectrum. To make better use of the available spectrum, cognitive radio (CR) is a smart technology that allows secondary licensees to borrow bandwidth from the principal licensee [1].

In order to alter transmission and reception, CR alters the parameters of communication, such as carrier frequency, bandwidth, and modulation techniques. When the cognitive radio network detects whether primary users are present or not, they are able to use licensed band resources more effectively. Different cognitive radio designs are feasible, but all of them necessitate the use of a spectrum sensing unit. The most important thing for spectrum sensing is the antenna selection, having an extremely wide bandwidth, dimension, high efficiency, and near isotropic radiation pattern [2,3].

For spectrum sensing applications, an octagonal antenna design with a bandwidth of 54.7% is presented in [4]. Recently, numerous antennas have been published in the literature for applications such as spectrum sensing and cognitive radio [5,6]. In ref [7] wide-band circular monopole antenna has been designed with an impedance bandwidth of 4.25–10 GHz. In [8] a printed super-wideband (SWB) antenna with an elliptical monopole and a trapezoid ground plane is designed to achieve high bandwidth.

Many antennas have recently been reported in the literature for spectrum sensing and cognitive radio applications [9,10,11,12,13,14,15,16].

As long as the antenna’s configuration is complementary to itself, the frequency-independent impedance property can theoretically be realised. To convert the input impedance from 188.5 Ω to 50 Ω, a matching network is required [17]. For ultra-wideband (UWB) applications, a bow-tie quasi-self-complementary (QSCA) and a castor leaf-shaped antenna with an average gain value of 4.36 dB and 80% radiation efficiency over its operating frequency range are presented in [18,19] and total gain variation of 0.5 dB in [20]. An elliptical mono-pole antenna with a triangular ground plane [21] covered a VSWR ≤ from 0.67 GHz to 12 GHz with consistent radiation patterns for spectrum sensing applications. In [22] reconfigurable narrow band micro strip antenna with a UWB dielectric resonator antenna (DRA) cover a bandwidth from 3–11 GHz and omnidirectional pattern provides 30 dB gain variation and 80% efficiency. A rectangular microstrip antenna with a spanner-shaped feeding line [23], printed hexagonal antenna [24] and elliptical mono-pole antenna [25] is presented for wireless application, cognitive radio application, and detection application respectively. The mono-pole antenna used in [26] is operational from 2.8–10.6 GHz achieved a gain variation of 60 dB. Planar monopole antennas with multi-port have a bandwidth from 2–11 GHz, a gain variation of 30 dB, and an efficiency of 50% for cognitive radio [27] and UWB mono-pole in [28] achieves 85% bandwidth and a gain variation of 25 dB for spectrum sensing. In [29] a boomerang-shaped UWB planar antenna has an impedance bandwidth of 163% with a variation of 40 dB gain.

An ellipse-shaped leaky antenna for an X-band will land the aircraft and drone in any direction during the war and high gain is achieved [30]. To mitigate, the interference between UWB and different narrow band different notches is introduced in the band through different techniques such as complementary split ring resonator (CSRR) and parasitic elements [31,32]. Combining unmanned aerial vehicles (UAV) with CR increases the general performance. The integration of CR with unmanned aerial vehicles (UAV) is presented in [33].

A detailed examination of these designs shows that researchers have put much effort into improving the antenna bandwidth by incorporating changes to a basic monopole or a microstrip patch antenna. Although the approach is simple, the antenna must be mostly oriented along Z-axis to have an omnidirectional pattern in the azimuth direction. This will limit the antenna’s capability for spectrum sensing as now the antenna has to be always placed vertically. Moreover, the dominant polarization of such an antenna may be linear/vertical, thereby missing horizontally polarized transmissions.

### 1.2. Motivation and Contribution

In this paper, the design of a miniaturized arc shaped near isotropic self-complementary antenna is proposed. In comparison to previously reported designs, the proposed antenna offers a wider impedance bandwidth, and miniaturization of total dimensions in comparison with the reference design. Furthermore, the antenna achieves the highest reported efficiency of 92% and a minimum efficiency greater than 50% in the impedance bandwidth of 2.4–18.4 GHz. In the 2.4–18.4 GHz frequency range, the antenna has a nearly isotropic emission pattern, making it an excellent option for CR spectrum sensing applications.CR is considered for key enabler for deploying technologies having high connectivity such as smart cities, 5G, the internet of things (IoT), and the internet of flying things (IoFT) such as UAV and drones. The isotropic performance of the antenna has been quantified by calculating the solid angle of the normalized gain. The proposed design achieves an impedance bandwidth of 146%.

### 1.3. Distribution of the Article

Our manuscript is divided into Six sections: antenna Configuration and Analysis are discussed in Section 2; simulation results are presented in Section 3. Section 4 presents the Experimental Results; Section 5 presents the Comparative analysis of the antenna while Section 6 concludes the paper.

## 2. Antenna Configuration and Analysis

An isotropic antenna requires to emit an entire sphere. It is well-known that a vertically placed dipole has uniform radiation in all phi directions. While a vertically placed dipole antenna has nulls in θ = 0 and θ = 180 direction. Therefore, for the antenna to radiate uniformly in all directions, there needs to be a mechanism to fill these nulls somehow. This can be achieved either by inserting additional radiators or by modifying the vertically placed dipole so that it starts radiating in other directions too. Firstly, a narrow band near isotropic planar antenna is designed as shown in Figure 1 [22]. Once maximum isotropic is achieved, then the design is modified to enhance the bandwidth. To miniaturized the size of the antenna, different techniques have been applied.The final design layout of our proposed design is shown in Figure 2. Table 1 lists the optimal parameters of proposed antenna. A λ/20 dipole is attached for impedance matching to stabilize the power pattern.

### 2.1. Near Isotropic Antenna

The design of a near isotropic antenna has been presented in [20]. For the interest of thoroughness, the design process is clearly illustrated in Figure 3, which depicts the patterns of various dipole components. The orthogonal nulls can be noticed in the horizontal and vertical segments. However, if they are merged directly, the entire pattern will still contain nulls. However, when the curved segment is added, the maximum isotropy is reached. A near isotropic performance is attained by adjusting the lengths of different segments.

### 2.2. Bandwidth Enhancement

It is well-known that for bandwidth enhancement of any antenna, the antenna geometry should be adjusted so that the antenna has numerous current paths of variable lengths where the main radiating dimension of the body is equal to a quarter of a wavelength. So different techniques applied initially use exponential function due to which the impedance matching improved somehow, and then the parameters are optimized for better matching performance. By optimization of antenna 1, we achieved high bandwidth up to 7 GHz, as shown in Figure 3b. Still, the problem is that their size was very large, i.e., 249 × 203 ${mm}^{2}$, and as we know spectrum sensing applications, a compact antenna is efficient. The next step is to reduce the size of the antenna to the desired application area.

### 2.3. Miniaturized the Design

In the Next step, by applying the self-complementary technique design size becomes more compact, as shown in Figure 3c. However, their $S_{11}$ parameter performance is not good due to impedance mismatching. While after using λ/20 dipole S11 parameter and the far-field pattern becomes improved, as shown in Figure 3d. Figure 2 and Figure 3d show the final and optimized design of the antenna. Optimally tuned values of the proposed design are presented in Table 1.

## 3. Results and Discussion

The planner arc-shaped near isotropic self-complementary antenna has been designed. First, a narrow bandwidth near the isotropic planar antenna is built, and then the design is adjusted to boost the bandwidth after miniaturising the design size. The geometry of the suggested antenna is illustrated in Figure 2, which is constructed on an FR−4 substrate ($\varepsilon_{r}$ = 4.4, h=1.6 mm). On one side of the substrate, horizontal, vertical, and arc dipole names $L_{arc}$, $S_{4}$, $S_{1}$, and $L_{s}$ are arranged. The antenna is directly connected to a 50 ohm coaxial cable without a matching circuit. To quantify the isotropic performance, a unique parameter, i.e., the antenna solid angle, has been used because the solid angle is the best method for isotropy performance.

### 3.1. Return Loss

The simulated return loss curves for the four antennas have been plotted versus frequency, as shown in Figure 3. The return loss result of the dipole antenna shows that it will only resonate at 1 GHz, as shown in Figure 3a. After applying bandwidth enhancement theory on the dipole antenna, improvement has been achieved from 1.3 GHz to 7 GHz, with a size of 249 × 203 mm, which is enormous. At the same time, dimension is the main issue for the spectrum-sensing antenna. Also, the power pattern is not near isotropic for many frequencies. When applying the self-complementary technique, the return loss improves. The proposed antenna covers the range of frequencies from 2.4 GHz to 18.4 GHz, which could be suitable for spectrum sensing applications because it almost covers the entire band of UWB.

### 3.2. Input Impedance

The input impedance for the proposed antenna has been simulated as depicted in Figure 4. Returned loss and input impedance have a significant correlation. There are many ways to describe this, but one example is that the return loss curve for an antenna suggested in this paper has a severe dip at around 6 GHz, which results in a real impedance of around 50 Ω, while the imaginary component moving closer to zero. Since the self-complementary approach has an overall response of 50 Ω, the real part tends toward that value while the imaginary part drops to 0 Ω inside that range. The simulated bandwidths are 146% (2.4–18.4 GHz) for VSWR ≤ as shown in Figure 4c.

### 3.3. Proposed Antenna Efficiency

The antenna Gain variation and efficiency are also in Table 2 and Figure 5, respectively. The radiation efficiency is used to relate the gain and directivity. The higher the efficiency, the lesser the heat loss, which means most power is radiated from the antenna. The radiation efficiency of the designed antenna approaches 92% at 3.5 GHz, oscillating between 57% and 92%, Which shows the proposed design supremacy over previously reported designs.

### 3.4. Surface Current Distribution

The current distribution for the proposed self-complementary antenna at 2.4, 6.5, 12.4 and 18.4 GHz is depicted in Figure 6. The current distribution shows that the antenna acts like a dipole in the resonant mode and provides near isotropic radiation patterns over a wide band.

### 3.5. Isotropy, Radiation Pattern and Solid Angle

An antenna’s pattern solid angle is represented by Equation (Equation 1)
(1)$$\Omega_{p}=\;\int \int \;P_{n}\left(\Theta ,\Phi \right)d\Omega$$
where $P_{n}$ represents the normalized radiation pattern. Radiation intensity or gain of the antenna in the cone of solid angle $\Omega_{p}$ may be assumed unity and will be equal to integration of normalized radiation pattern over 4π steradian solid angle. The solid angle of the normalized radiation pattern, although never used before, can be a good method to quantify the isotropic performance of an antenna. The isotropic performance of the antenna at different frequencies will therefore be quantitatively evaluated by computing the solid angle of normalized radiation pattern at those frequencies. in the context of solid angle i.e., higher the value of solid angle, the better the isotropy. It is also worth mentioning here that since solid angle has not been previously used for quantification of isotropic performance, antenna radiation patterns have also been presented to validate the isotropic performance. The proposed antenna radiation patterns at different frequencies, i.e., 2 GHz, 3 GHz, 3.5 GHz, 3.6 GHz, 4 GHz, 5.5 GHz, 7 GHz, 8.5 GHz, and 15.6 GHz, with and without λ/20 dipole effects are shown in Figure 7 and Figure 8. There is a great effect of the λ/20 dipole on the radiation pattern. Without the dipole, the pattern is omni-direction for all frequencies. While using the dipole, a near-isotropic antenna pattern has almost been maintained.The solid angle for different standard antennas in comparison to the proposed antenna is shown in Figure 9. It can be seen that the solid angle of the proposed antenna have similar values as that of the standard antenna candidates.So proposed antenna achive high value of solid angle which is near to 3. Moreover, it can also be seen that lower value of solid angle at higher frequencies, as shown in Figure 9, is actually due to the presence of more frequent nulls within the radiation pattern thereby bringing the overall value of solid angle down.

## 4. Theoretical Experimental Results

A prototype of the proposed antenna is printed on FR−4 substrate ($\varepsilon_{r}$ = 4.4, h=1.6 mm) and simulated on CST. The prototype is tested in the chamber and E-H radiation patterns for co and cross-polarization are measured for different frequencies 2.5 GHz, 3.5 GHz, 6.5 GHz, and 17 GHz. The simulated and measured results for co and cross-polarization are in good comparison the results are shown in Figure 10 and Figure 11. The prototype is shown in Figure 12. A coaxial cable is used to feed the antenna. An agilent E5071B network analyzer is used in the measurement. Figure 13 shows the simulated and measured reflection coefficient. The simulated and measured results of the fabricated antenna with λ/20 agree very well.

From 2.4 to 18.4 GHz, the return loss value is less than −10 dB. The result demonstrates that the λ/20 improves the antenna matching and power pattern and supports the previously presented numerical results. As shown in Figure 7 and Figure 8, the overall radiation pattern in the YZ plane is fairly stable over the whole frequency range. It is further supported by the solid angle measurement displayed in Figure 9 for normalized gain. Very good stability is seen over the matched frequency bands, and a nearly isotropic pattern is produced.

## 5. Comparative Analysis of Antenna

Antenna performance against previously reported designs is also compared and is presented in Table 2. Performance has been compared against electrical Size, bandwidth, efficiency (Min., Max), gain variation and peak gain (dB). It is important to note that gain variation at this point means the difference between the 0 dB normalized value and the deepest Null point. Compared to the antennas presented in the literature, as shown in Table 2, the proposed antenna has a better efficiency performance, gain variation, and small size compared to the reported antenna, making them most suitable for spectrum sensing applications.

## 6. Conclusions

In this paper, the design and testing of an arc-shaped self-complementary antenna are described.An arc-shaped dipole with horizontal and vertical arms is used to achieve a Near isotropic radiation pattern. Simulated and experimental results show that the proposed antenna has an impedance bandwidth of 146% (2.4–18.4 GHz) for VSWR ≤ with a good radiation pattern. In order to quantify the antenna performance, antenna Gain variation, bandwidth, efficiency, and size have been compared with previously reported designs. It is shown that the proposed arc-shaped antenna can achieve nearly isotropic radiation patterns with a maximum radiation efficiency of 92%. There is a good match between the simulated and measured data, making them appropriate for spectrum sensing and high connectivity technology such as 5G, IoT, IoFT, and drones.

## Figures and Tables

**Figure 1 sensors-23-00927-f001:**
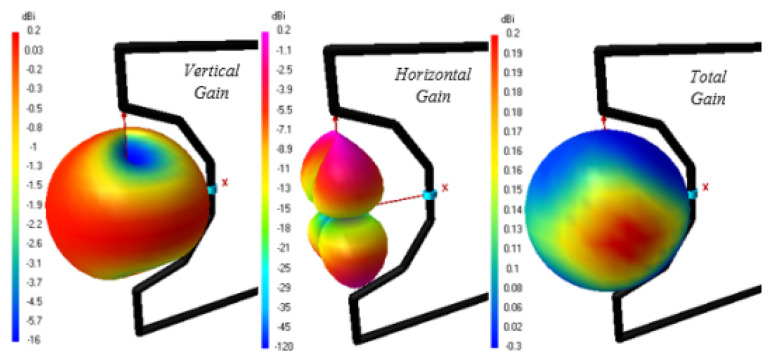
Structure of Near Isotropic Antenna.

**Figure 2 sensors-23-00927-f002:**
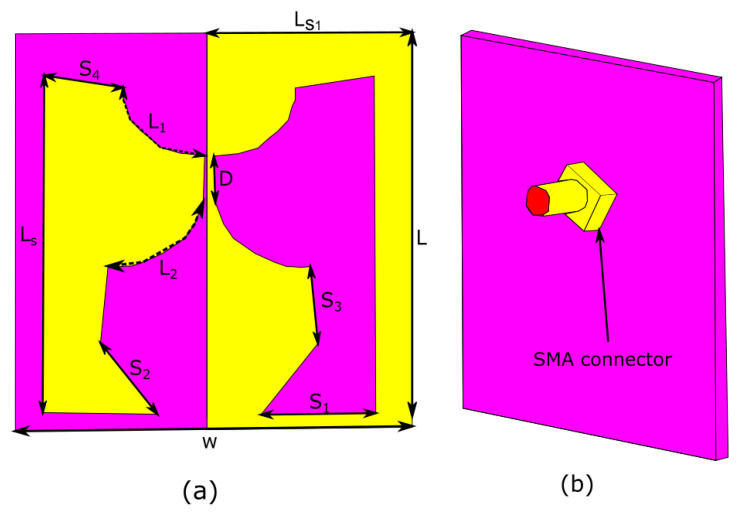
Schematic view of the proposed arc shape self-complementary antenna. (**a**) Top view of the proposed antenna (**b**) Back view of the proposed antenna.

**Figure 3 sensors-23-00927-f003:**
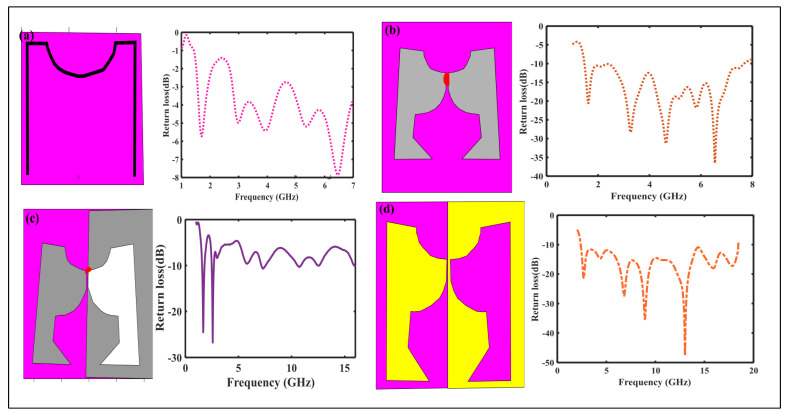
Steps for Proposed Antenna Design (**a**) arc shape dipole antenna (**b**) a boomerang-shaped UWB antenna (**c**) By applying Self-complementary technique (**d**) optimized self-complementary shaped of dipole.

**Figure 4 sensors-23-00927-f004:**
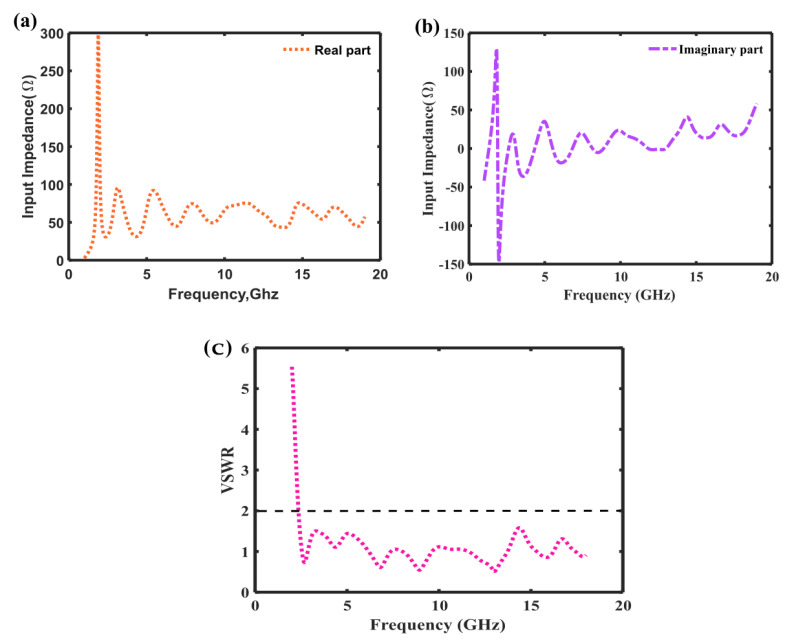
(**a**) Input Impedance (Real Part) (**b**) Input Impedance (Imaginary Part) (**c**) VSWR.

**Figure 5 sensors-23-00927-f005:**
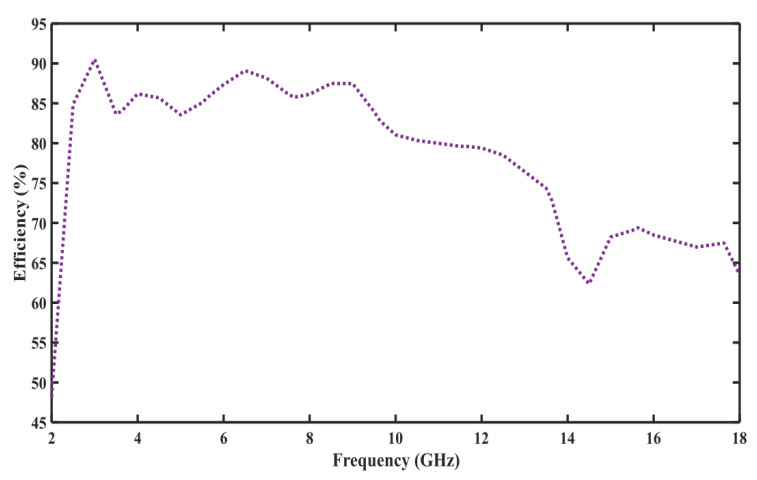
Efficiency of the Proposed Antenna.

**Figure 6 sensors-23-00927-f006:**
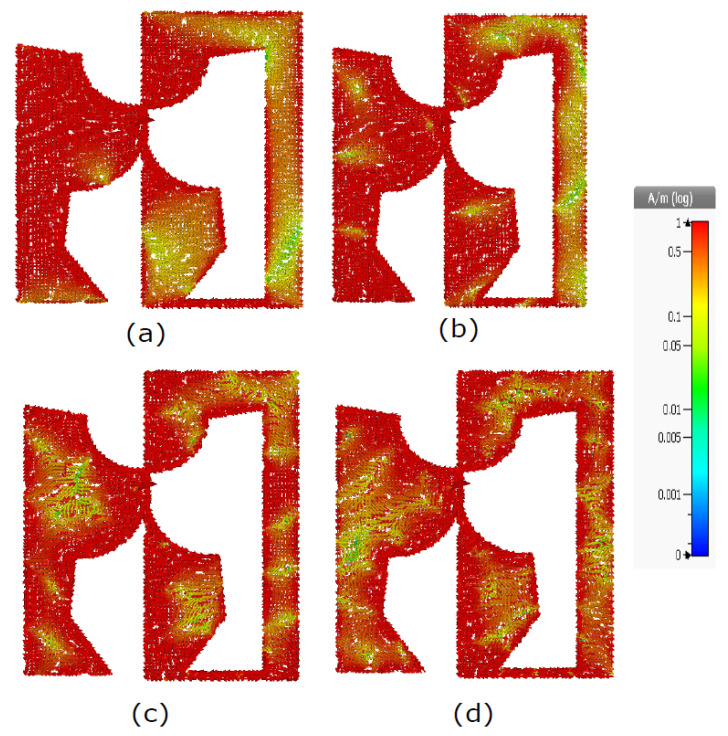
Simulated Surface Current Distribution at (**a**) 2.4 GHz (**b**) 6.5 GHz (**c**) 12.4 GHz and (**d**) 18.4 GHz.

**Figure 7 sensors-23-00927-f007:**
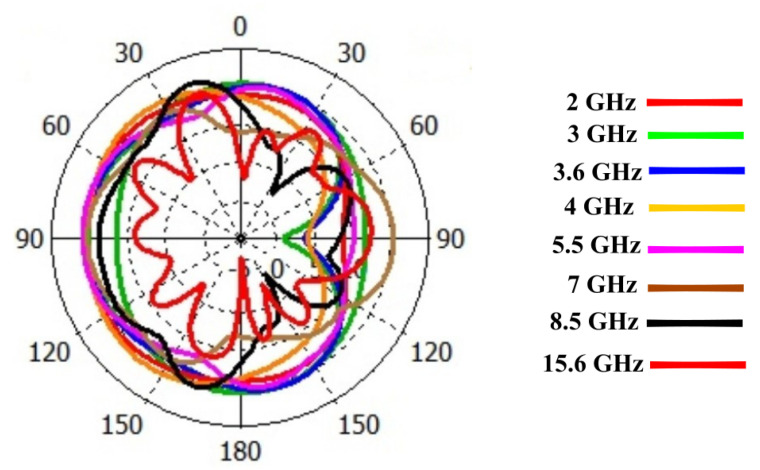
Power Pattern (YZ-plane) without the λ/20 Dipole.

**Figure 8 sensors-23-00927-f008:**
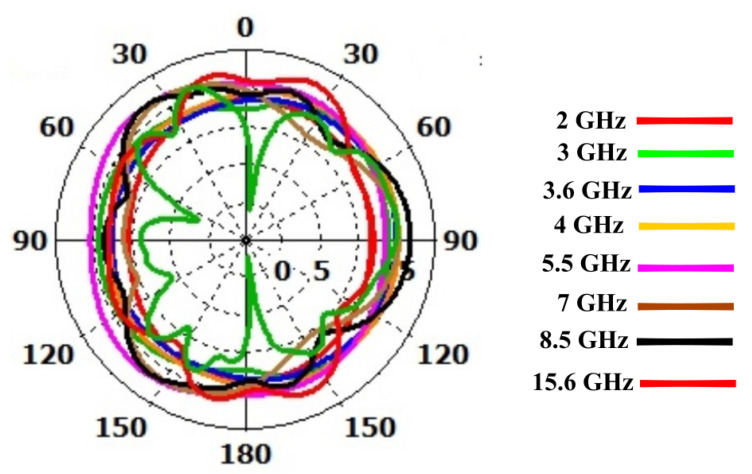
Power Pattern (YZ-plane) with the λ/20 Dipole.

**Figure 9 sensors-23-00927-f009:**
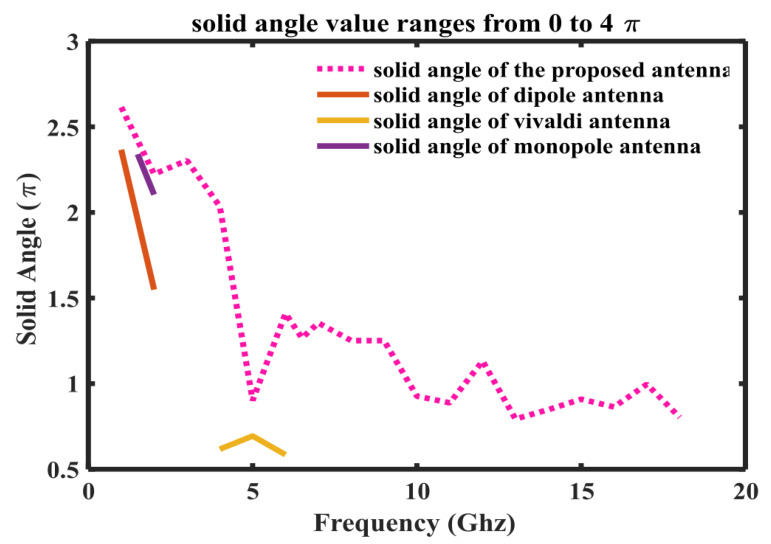
Solid angle of different antenna.

**Figure 10 sensors-23-00927-f010:**
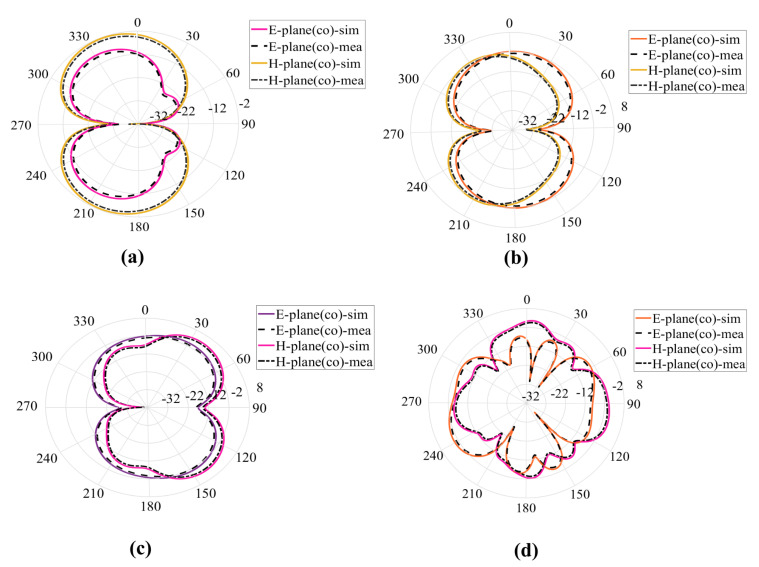
E-H plane radiation patterns for Co-polarization for different frequencies (**a**) 2.5 GHz (**b**) 3.5 GHz (**c**) 6.5 GHz (**d**) 17 GHz.

**Figure 11 sensors-23-00927-f011:**
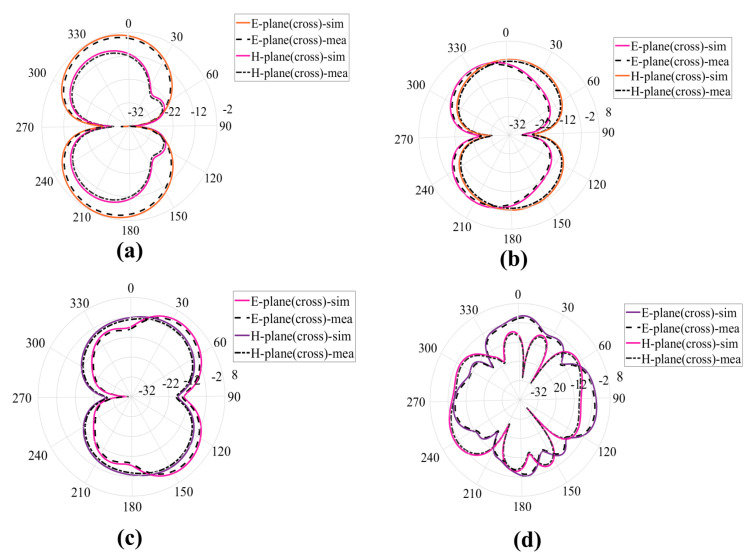
E-H plane radiation patterns for Cross-polarization for different frequencies (**a**) 2.5 GHz (**b**) 3.5 GHz (**c**) 6.5 GHz (**d**) 17 GHz.

**Figure 12 sensors-23-00927-f012:**
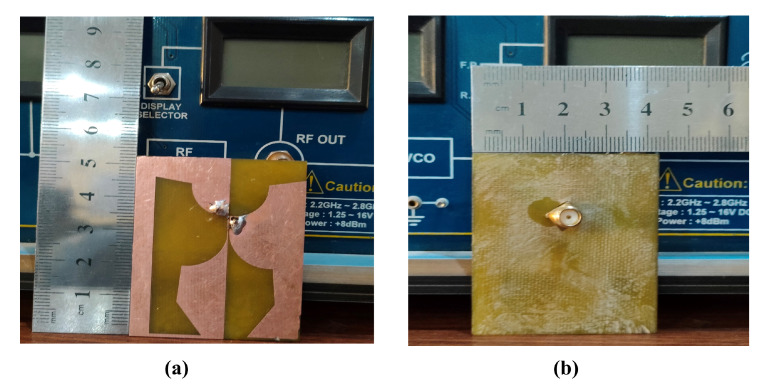
Prototype of the Proposed Antenna (**a**) top view (**b**) bottom view.

**Figure 13 sensors-23-00927-f013:**
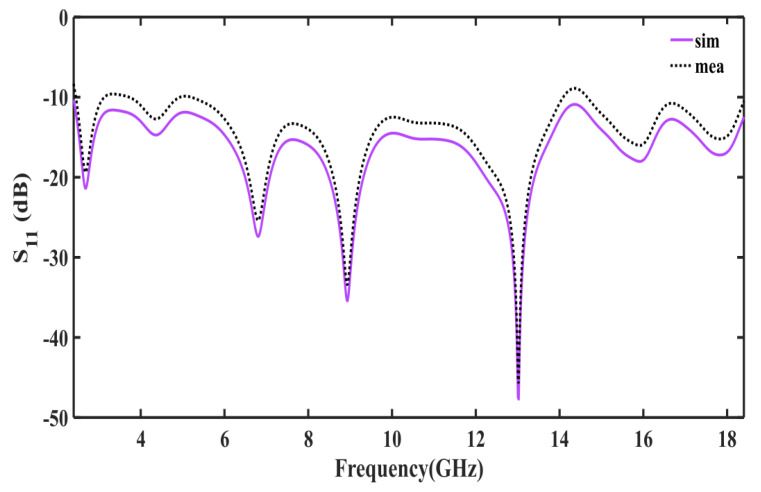
Measured and simulated $S_{11}$ of the proposed antenna.

**Table 1 sensors-23-00927-t001:** Proposed Antenna Dimensions.

Parameter	Dimension (mm)	Parameter	Dimension (mm)
$S_{1}$	14	$L_{S}$	47.88
$S_{2}$	12.2	$L_{arc}$	13.87
$S_{3}$	11	D	6.69
$S_{4}$	10	$L_{s1}$	25.15
L	55	W	45
$L_{1}$	13.87	$L_{2}$	16.92

**Table 2 sensors-23-00927-t002:** Comparison of proposed antenna with existing designs.

Ref.	Size ($\lambda^{2}$)	BW	Gain Var.	Eff. (Min.–Max.)	Peak Gain
[21]	0.4 × 0.36	107%	−40 dB	-	-
[22]	0.65 × 0.4	114%	−30 dB	80–90%	5 dB
[23]	0.24 × 0.16	153.22%	−45 dB	65–85%	5.1 dB
[24]	0.5 × 0.5	108.64%	−30 dB	-	7 dB
[25]	0.43 × 0.315	143%	−40 dB	-	5.5 dB
[26]	0.37 × 0.37	116%	−60 dB	-	-
[27]	0.53 × 0.266	138%	−30 dB	50–86%	5.5 dB
[28]	0.74 × 1.42	85%	−25 dB	-	4.4 dB
[29]	0.83 × 0.85	163%	−35 dB	55–92%	-
**Proposed**	**0.44 × 0.36**	**146%**	**−40 dB**	**52–91%**	**4.5 dB**

## Data Availability

Not applicable.

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
