# Peer review of "A Miniaturized Arc Shaped Near Isotropic Self-Complementary Antenna for Spectrum Sensing Applications"

_sensors, 2023, doi:10.3390/s23020927_

Round 1
Reviewer 1 Report
please see the attachment

Author Response
Response to Reviewer 1 Comments
We are thankful to the Editor for giving us the opportunity to revise and resubmit the manuscript. We are also grateful to the worthy reviewers for their valuable comments which helped to improve quality of the manuscript. We have introduced our responses to the reviewers’ comments below. The text in black is reviewer’s comment and in blue italic is authors’ response.
Reviewer: 1
This paper presents the design of an arc-shaped near-isotropic self-complementary antenna for spectrum sensing applications. Based on the results shown in this manuscript, the measurement method was less original, and the quality of the manuscript needs to improve seriously which can be up to the Journal’s standard. I summarized some suggestions to the author for improving the quality of the manuscript.
We are thankful to the worthy reviewer for reviewing the paper and appreciating our work. We are also thankful to the worthy reviewer for giving valuable suggestions which helped in improving the quality of our work.
Comment-1:
The simulated design and fabricated unit are not represented properly. I think that can’t be accepted for publishing according to the policy of Sensors. The experimental results are not presented properly.
Answer:
The simulated unit is represented in Figure 2 with proper dimension and unit and the fabricated unit is represented in Figure 12. The front sideshows the complementary antenna and back side show the excitation of the design.
Figure : Simulated unit (a) front side (b)back side excitation
Fabricated unit (c)Front side (b)Back side
In the revised manuscript we added the clear figure.
Comment-2:
In the abstract 3rd line N should be a small letter.
Answer:
We corrected all the grammatical, punctuation, and typo mistakes in the revised manuscript.
Comment-3:
In the introduction, line number 23 reference has not been added (and distributes it amongst the licenced users[? ]). Please correct this
Answer:
In the revised manuscript, we corrected it.
Comment-4:
As per [4], For spectrum sensing applications, it achieved a bandwidth of 190.35%. But your antenna is only 146%. Then how your antenna is better than this antenna?
Answer:
The achieved bandwidth in reference [4] is 54.7% not 190.35% it’s a typo mistake and in the revised manuscript, we have corrected the grammatical and punctuation mistake.
Comment-5:
Please mention what is the bandwidth in [18] and compare it with your design.
Answer:
There is impedance bandwidth of reference [18] is 120% (3–12 GHz). But the reference [18] antenna is not design for spectrum sensing application. As we know, the most important challenges for spectrum sensing antennas include their ability to operate within an extremely wide bandwidth, their efficiency, and dimensions, and the antenna has an isotropic radiation pattern so that active transmission can be sensed, within the band of operation, without worrying about the direction of arrival as mentioned in [1], [2], [3] references of the manuscript.
But reference [18] don’t full fill other requirements except the wide bandwidth which is also less then of our proposed design so no need of comparison with our design. Our proposed design fulfills all these requirements. when comparing our antenna with the reported antenna in literature, as shown in table 2, our antenna's performance is good in every aspect of view. So our proposed design is most suitable for spectrum sensing applications
Comment-6:
No continuity from line number 39 to 40. Why suddenly TCM from the previous paragraph? Also, no continuity from line number 40 to 44 paragraph. This work TCM is not used and why this paragraph. In my opinion, this paragraph is completely un relevant to the proposed design. Un relevant references cited. Please exclude these.
.
Answer:
We updated the introduction section, removed the irrelevant paragraph and citation, and added more relevant references [9-16].
Comment-7:
No uniformity in citing references in the introduction. For example, some references mentioned BW and superior designs not mentioned bandwidth.
Answer:
We updated the introduction section, removed the irrelevant paragraph and citation, and added more relevant references [9-16].
Comment-8:
There is no continuity in the introduction. It needs to be rewritten.
Answer:
We updated the introduction section, removed the irrelevant paragraph and citation, and added more relevant references [9-16].
Comment-9:
What are SCA and QSCA.
Answer:
SCA stand for ‘’self-complementary antenna ‘’ and QSCA stand for quasi self-complementary antenna’’.
Comment-10:
Are the compared references belonging to self-complementary?
Answer:
No, because till now, no self-complementary antenna has been designed for spectrum sensing applications. Recently only one conference paper has been published, which we have in the revised manuscript as a reference [15].
Comment-11:
In [23] minimum efficiency is greater than your design. Also, gain and dimensions are superior to your design along with bandwidth. How the authors will justify your design is better than this by only upper efficiency of 92%.
Answer:
The isotropic radiation pattern is an essential parameter for spectrum sensing applications because the antenna has an isotropic radiation pattern so that active transmission can be sensed, within the band of operation, without worrying about the direction of arrival, as mentioned in [1], [2], [3] references of the manuscript. So, our focus is not only on wide bandwidth we have to also focus on radiation patterns and other parameters.
In the comparison table, we included gain variation (Min., Max) and peak gain (dB), which is an important parameter to measure the isotropic performance of any antenna. It is important to note that gain variation at this point means the difference between the 0 dB normalized value and the deepest Null point. If we look into the most profound null value of ref [23], it is 45 dB, which is more than the proposed design because the isotropy performance is not good. Also, the proposed antenna's maximum efficiency is 92%, while ref [23] has a maximum efficiency of 85%, which makes our design better than the ref [23] design.
Comment-12:
The authors have not followed uniformity in drawing the figures.
Answer:
In the revised manuscript we have redrawn all the figure and make it more clear.
Comment-13:
“Our manuscript is divided into Six sections: antenna Configuration and Analysis are discussed in Section II; simulation results are presented in Section III. section IV presents the Experimental Results; Section V presents the Comparative analysis of the antenna while Section VI concludes the paper”. But here sections are given numerical values. it needs to be corrected.
Answer:
The revised manuscript is updated.
Comment-14:
The results description is not up to the mark of SCI journals. No uniformity in the results. Some results are mentioned in color and some are in black and white. More results are directly taken a screenshot from the simulation.
Answer:
We have redrawn all the figures in the revised manuscript and made them more apparent.
Comment-15:
As from the impedance chart, it is mentioned around 50 Ohm in the frequency range. But most of the frequency range impedance is far away from 50 Ohms.
Answer:
When we used discrete in simulation, we got good results because there was no loss, but when we used the SMA connector in simulation, our impedance moved away from 50 Ohm due to the losses of the SMA connector. But it is acceptable, as mentioned in ref [15]. As in the revised manuscript, we included the VSWR graph, which covered a VSWR≤2 from 2.4 GHz to 18.4 for the whole frequency band, which is the acceptable range.
Comment-16:
Above 10 GHz efficiency is degraded to below 85% then how it is feasible to lesser the heat loss.
Answer:
Getting near to isotropic performance in such high bandwidth is difficult as we can see a maximum antenna that achieves an isotropic pattern is just for a single narrow band, and an efficiency greater than 50% is acceptable for spectrum sensing, as mentioned in ref [1].
Comment-17:
By analyzing how the authors have understood the following statement” provides near isotropic radiation patterns over a wide band”.
Answer:
An isotropic antenna is a theoretical antenna that radiates equally in all directions - horizontally and vertically with the same intensity. Radiation intensity or gain of the antenna in the cone of solid angle may be assumed unity. It will equal the integration of normalized radiation patterns over 4π steradian solid angle. Now we can see the solid angle of the proposed at a lower frequency, almost similar to 3π Moreover. It can also be seen that the lower value of solid angle at higher frequencies, as shown in figure 9, is due to the presence of more frequent nulls within the radiation pattern, thereby bringing the overall value of solid angle down. It is important to note that gain variation at this point means the difference between the 0 dB normalized value and the deepest Null point. So overall, if we look into the antenna radiation pattern and compare it with the perfect isotropic antenna in table 2, the radiation pattern is near to isotropic.
Comment-18:
“are shown in Figure 7 Figure 8, respectively” need to be corrected.
Answer:
We have redrawn all the figures in the revised manuscript and made them more apparent.
Comment-19:
The authors claimed that it is isotropic, but the E-plane and H-Plane pattern are not showing the isotropic pattern.
.
Answer:
The 3D radiation pattern is shown in Re-fig1.its can be seen that at the lower frequency, the radiation pattern is near to isotropic having a null, while at a higher frequency, the deepest null is due to which the isotropy value is down, but if we look at the overall pattern its look like near isotropic not omnidirectional.
. It can also be seen that the lower value of solid angle at higher frequencies, as shown in figure 9, is due to the presence of more frequent nulls within the radiation pattern, thereby bringing the overall value of solid angle down. It is important to note that gain variation at this point means the difference between the 0 dB normalized value and the deepest Null point. So overall, if we look into the antenna radiation pattern and compare it with the perfect isotropic antenna in table 2, the radiation pattern is near to isotropic.
Re-Fig1:3D radiation pattern at (a) 3 GHz (b) 3.5 GHz (c) 14.5 GHz (d) 17 GHz
We are thankful to Reviewer 1 for the important and helpful comments which has significantly improved the quality of our work.
Thanks, and Best Regards
Authors
06-01-2023

Reviewer 2 Report
The authors present an arc shaped dipole antenna for spectrum sensing applications. The antenna size, bandwidth and gain results are interesting.
Please find my comments to improve the paper quality:
1- Please could you explain in line 8 what it means "good" radiation pattern?
2- The authors present the term solid angles early in the paper, then explain it later. My proposal to move the section 3.5, and put it in
Section 2, as this aspect is crucial to validate the antenna performance.
3- Please could you explain why the antenna matching results are degraded in Fig.3 c after applying the Self-complementary technique?
3- Please could you provide the loss tag or Tang Delta of the used FR-4 substrate?
4- Please could you improve the figures 3 and 4 quality (font size and line width)? Also other figures resolutions are poor.
5- Could you explain the high levels of cross-polarization radiation patterns, especially at lower frequencies?
6- Please could you add Ref [30] to table 2 for comparison of your design?
7- Please could you provide the gain curve versus frequencies?
Please check the paper format, and English grammar.
1- In abstract:
line 5: Space before "An".
Line 6 Near "N" miniscule.
2- Reference problem in line 23.
3- Line 36, "ref [7]" presented in wrong format.
4- Line 146 "when" W is capital.
Author Response
Response to Reviewer 2 Comments
We are thankful to the Editor for giving us the opportunity to revise and resubmit the manuscript. We are also grateful to the worthy reviewers for their valuable comments which helped to improve quality of the manuscript. We have introduced our responses to the reviewers’ comments below. The text in black is reviewer’s comment and in blue italic is authors’ response.
Reviewer: 2
the authors present an arc shaped dipole antenna for spectrum sensing applications. The antenna size, bandwidth and gain results are interesting. Please find my comments to improve the paper quality:
We are thankful to the worthy reviewer for reviewing the paper and appreciating our work. We are also thankful to the worthy reviewer for giving valuable suggestions which helped in improving the quality of our work.
Comment-1:
Please could you explain in line 8 what it means "good" radiation pattern?
Answer:
An isotropic antenna is a theoretical antenna that radiates equally in all directions - horizontally and vertically with the same intensity. Radiation intensity or gain of the antenna in the cone of solid angle may be assumed unity and will be equal to integration of normalized radiation pattern over 4π steradian solid angle. Now we can see the solid angle at lower frequency almost equal to 3π Moreover, it can also be seen that lower value of solid angle at higher frequencies, as shown in figure 9, is due to the presence of more frequent nulls within the radiation pattern thereby bringing the overall value of solid angle down.so overall if we look into the radiation pattern of antenna and compare it with isotropic antenna its good.
Comment-2:
The authors present the term solid angles early in the paper, then explain it later. My proposal to move section 3.5, and put it in Section 2, as this aspect is crucial to validate the antenna performance.
Answer:
Thanks for reviewing carefully. In section 2 we discussed the antenna configuration and analysis. This section explains the near isotropic, bandwidth enhancement, and miniaturization of design. Section 3 explains the results and discussion. This section implies return loss, input impedance, efficiency, current distribution, and isotropy radiation pattern, and solid angle.
Comment-3:
Please could you explain why the antenna matching results are degraded in Fig.3 c after applying the Self-complementary technique?
Answer:
When we directly applied the self-complementary technique to antenna 2, their size was enormous, and there was an impedance mismatch. By optimizing the antenna size, we got the required results.
Comment-4:
Please could you provide the loss tag or Tang Delta of the used FR-4 substrate?
Answer:
The loss tangent of FR-4 is 0.02. We also included it in the revised manuscript.
Comment-5:
Please could you improve the figures 3 and 4 quality (font size and line width)? Also, other figures resolutions are poor.
Answer:
We redraw all the figures in the revised manuscript and make them explicit.
Comment-6:
Could you explain the high levels of cross-polarization radiation patterns, especially at lower frequencies?
Answer:
At lower frequency, the cross-polarization is good, and the level of null is two, but for high frequency, the number of most profound null increase the overall isotropy performance decrease..
Comment-7:
Please could you add Ref [30] to table 2 for comparison of your design?
Answer:
As we know, the most important challenges for spectrum sensing antennas include their ability to operate within an extremely wide bandwidth, their efficiency, and dimensions, and the antenna has an isotropic radiation pattern so that active transmission can be sensed, within the band of operation, without worrying about the direction of arrival as mentioned in [1], [2], [3] references of the manuscript. So, our focus is not only on wide bandwidth we have to also focus on radiation patterns and other parameters. Initially, we designed a single narrow-band antenna; we started working on increasing its bandwidth by applying different techniques and then miniaturizing the size to get the required antenna for spectrum sensing. While reference [30] is not an isotropic antenna and is not for spectrum sensing applications, we have not put it in the comparison table. We put the antenna in a comparison table that achieved all these properties.
Comment-8:
Please could you provide the gain curve versus frequencies?
Answer:
Actually, in the comparison table, we included gain variation (Min., Max) and peak gain (dB), .which is an important parameter to measure the isotropic performance of any antenna. It is important to note that gain variation at this point means the difference between the 0 dB normalized value and the deepest Null point. But in the response letter, I also include the Frequency vs. Gain graph.
Comment-9:
Please check the paper format, and English grammar.
1- In abstract:
line 5: Space before "An".
Line 6 Near "N" miniscule.
2- Reference problem in line 23.
3- Line 36, "ref [7]" presented in wrong format.
4- Line 146 "when" W is capital.
Answer:
The manuscript is revised and free from grammatical, typos, and wrong format presentation.
We are thankful to Reviewer 2 for the important and helpful comments which has significantly improved the quality of our work.
Thanks, and Best Regards
Authors
06-01-2023

Round 2
Reviewer 1 Report
The authors addressed all my comments and improved the revised version